# Chemical genomics reveals histone deacetylases are required for core regulatory transcription

Berkley E. Gryder [1,8], Lei Wu[2,8], Girma M. Woldemichael[3], Silvia Pomella[1,4], Taylor R. Quinn[5], Paul M.C. Park [2], Abigail Cleveland[1], Benjamin Z. Stanton [1], Young Song[1], Rossella Rota [4], Olaf Wiest[5], Marielle E. Yohe[6], Jack F. Shern[6], Jun Qi[2,7] & Javed Khan [1]

Identity determining transcription factors (TFs), or core regulatory (CR) TFs, are governed by cell-type specific super enhancers (SEs). Drugs to selectively inhibit CR circuitry are of high interest for cancer treatment. In alveolar rhabdomyosarcoma, PAX3-FOXO1 activates SEs to induce the expression of other CR TFs, providing a model system for studying cancer cell addiction to CR transcription. Using chemical genetics, the systematic screening of chemical matter for a biological outcome, here we report on a screen for epigenetic chemical probes able to distinguish between SE-driven transcription and constitutive transcription. We find that chemical probes along the acetylation-axis, and not the methylation-axis, selectively disrupt CR transcription. Additionally, we find that histone deacetylases (HDACs) are essential for CR TF transcription. We further dissect the contribution of HDAC isoforms using selective inhibitors, including the newly developed selective HDAC3 inhibitor LW3. We show HDAC1/2/3 are the co-essential isoforms that when co-inhibited halt CR transcription, making CR TF sites hyper-accessible and disrupting chromatin looping.

[1] Genetics Branch, NCI, NIH, Bethesda, MD 20892, USA. [2] Department of Cancer Biology, Dana-Farber Cancer Institute, Boston, MA 02215, USA. [3] Molecular Targets Laboratory, Frederick National Laboratory for Cancer Research, Frederick, MD 21701, USA. [4] Department of Oncohematology, Laboratory of Angiogenesis, Ospedale Pediatrico Bambino Gesu' Research Institute, Rome 00165, Italy. [5] Department of Chemistry & Biochemistry, University of Notre Dame, Notre Dame, IN 46556, USA. [6] Pediatric Oncology Branch, CCR, NCI, NIH, Bethesda, MD 20814, USA. [7] Department of Medicine, Harvard Medical School, Boston, MA 02115, USA. [8] These authors contributed equally: Berkley E. Gryder, Lei Wu. Correspondence and requests for materials should be addressed to J.Q. (email: jun_qi@dfci.harvard.edu) or to J.K. (email: khanjav@mail.nih.gov)

The traditional perception of transcription factors (TFs) as undruggable has been overturned by strategies which employ chemical probes[1] to halt the epigenetic apparatus that TFs co-opt for transcription. Selective inhibition of transcription of MYC and other key oncogenic TFs has been achieved by inhibition of acetyl-lysine reader (bromodomain)-containing proteins[2–5]. In pediatric rhabdomyosarcoma, for example, BET bromodomain inhibition selectively ablates not only the transcription of MYC and MYCN, but also lineage-specific TFs (that become oncogenic in the disease context) such as MYOD1[6]. While most TFs are expressed at low levels and are not particularly attractive drug targets, a small set of super enhancer (SE)-driven TFs form strong autoregulatory circuitry, the core regulatory (CR) TFs, and are essential for cancer growth. Often pan-cancer oncogenes, such as the MYC family of proteins, form a part of the CR circuitry[6,7], in addition to lineage-specific TFs inherited from the cell of origin[8,9].

Here, we sought to identify small molecules capable of selectively disabling CR circuitry, using PAX3-FOXO1 fusion oncogene positive rhabdomyosarcoma (FP-RMS) as a model system. Using both large agnostic screening (63,000 compounds) and 77 mechanistically curated epigenetic and transcriptional probes, we report SE-driven transcription has a rapid and selective dependence on readers, writers, and erasers of histone acetylation, while small molecule modulators of histone methylation have almost no impact within a 24 h window. RNA-seq screening further confirmed that acetylation-axis-perturbing probes selectively ablate transcription of CR networks, while methylation-axis probes do not. Bromodomains, which assemble to the genome by binding the acetyl-lysine histone scaffold associated with active enhancers and promoters, are essential for CR TF-dependent transcription[2]. Surprisingly, we discovered histone deacetylase (HDAC) enzymes which remove acetylation are also essential for CR transcription, exposing a new mechanism underlying the long appreciated phenotypic consequences of chemical probes inhibiting the enzymatic activity of various HDAC isoforms. We utilized a set of HDAC selective chemical probes, including a newly developed HDAC3 selective inhibitor, to dissect the contributions of zinc-dependent HDACs to CR TF transcriptional control. We uncovered that nuclear Class I HDACs 1, 2, and 3 but not HDAC8 are co-essential for CR transcription, and simultaneous inhibition of HDAC1/2/3 disrupts CR TF chromatin architecture.

## Results

**Chemical probes expose SE vulnerabilities.** CR TFs bind in a combinatorial fashion to create large protein aggregates at the interface of a gene promoter and usually several long-distance regulatory elements, forming a "super enhancer", which are now known to form large multivalent phase condensates. We reasoned that the unique chemical features of the relatively few genes which are controlled in this way would transfer to unique responses to chemical probe perturbation.

With the goal of characterizing small molecule vulnerabilities of CR TF-driven gene expression, we used our previously reported engineered cell line with a luciferase reporter driven by an intronic SE within the ALK gene locus, one of the most subtype selective cis-regulatory elements in FP-RMS[6] (Fig. 1a). We previously confirmed that this system specifically reports the activity of PAX3-FOXO1 in FP-RMS and is silent in FN-RMS cells[6]. A constitutively active, promoter-driven luciferase was used as a control. Both are lentiviral, chromatin-incorporated reporters. Chromatin immunoprecipitation followed by sequencing (ChIP-seq) showed that this cis-regulatory element was heavily occupied by all members of the CR circuitry of FP-RMS (SOX8, MYOG, MYOD1, MYCN, PAX3-FOXO1) along with

epigenetic components of SE-mediated gene activation (such as p300, Mediator, and BRD4, Fig. 1a). Chemical probe-based Chem-seq with bio-JQ1 affirmed the chemical targeting of BRD4[10] in the context of this regulatory locus (Fig. 1a), and selective sensitivity of the ALK SE to JQ1 was shown by our lab to be an effective in vivo metric of on-target efficacy for SE-driven transcription[6].

We then employed a set of chemical probes targeting epigenetic regulatory complexes (i.e., targeting chromatin reader, writer, and eraser proteins) that can directly perturb acetylation and/or methylation states of histone tails[11], in addition to inhibitors of diverse steps in transcription of mRNA to identify the epigenetic proteins required for CR TF function. These inhibitors were assayed for the dose responsiveness of a constitutive CMV promoter driven and the SE-driven luciferase. Responses were considered inactive if >60% of the SE luciferase remained; SE inhibition responses were then divided into three categories based on comparison to CMV response: general transcriptional inhibition (SE ≤ 60%, ≤30% difference between SE and CMV), SE selective (SE ≤ 60%, CMV ≤ 170%), and SE down CMV up (SE ≤ 60%, CMV > 170%). General transcriptional inhibition was seen with triptolide-inhibiting XPB of TFIIH (Pol2 initiation), α-Amanitin inhibiting the Pol2 trigger loop, flavopiridol-inhibiting CDK9 in the context of positive transcription elongation factor (pTEF-b), while a subtle SE-selectivity for CDK7 inhibition by THZ1 (Fig. 1b) was consistent with previous observations[12]. The relative selectivity of SE-driven luciferase was validated with direct BET bromodomain inhibition induced by several BET inhibitors, such as JQ1, that possess remarkable enhancer sensitivity compared to constitutive transcription (Fig. 1b).

In addition to inhibitors of the enhancer-associated chromatin factor BRD4, we discovered histone-acetylation writer proteins (p300/CBP) and eraser enzymes (HDACs) were especially required for enhancer-driven but not promoter-driven luciferase induction (Fig. 1c, Supplementary Fig. 1). We generally found that epigenetic chemical probes effecting lysine methylation states (writers, erasers, or readers) or DNA methylation had very little selectivity, nor did signaling inhibitors (Fig. 1c, Supplementary Fig. 1). HDAC inhibitors, well known for their ability to increase expression of many genes, caused an increase in CMV induction, while at the same doses caused complete disruption of SE output (Fig. 1d). This HDAC inhibition of SE activity was also disrupted prior to cell death, whereas transcriptionally unselective inhibitors were only able to shutdown SE activity at concentrations causing cell death (cf., LSD1 inhibitor LSD690, Fig. 1e) suggesting an indirect effect.

**Large-scale transcriptional screen for SE-selectivity.** Upon validating the ability of our cell-based system to rank small molecules for selective SE transcriptional impact, we extended it to a library of largely uncharacterized pure-compounds. Each compound of this 63,000-member chemical library was tested at a final concentration of 10 µM, in both cells harboring CMV-promoter and SE-driven luciferase, which were measured at 24 h after compound exposure (Fig. 2a). We performed follow-up dose–response evaluations for 573 of the most selective or potent molecules (Fig. 2b). We classified compounds into either inactive, SE selective (reduction in SE transcription without effecting promoter-driven transcription), SE inhibitory while promoter upregulating, or non-selectively active (Fig. 2b). While most molecules originally identified at 10 µM did not continue to potently and selectively inhibit SE-driven transcription at lower doses, N1302 (1-alaninechlamydocin) was able to inhibit SE activity at concentrations as low as 2 nM (Fig. 2b). N1302 is an

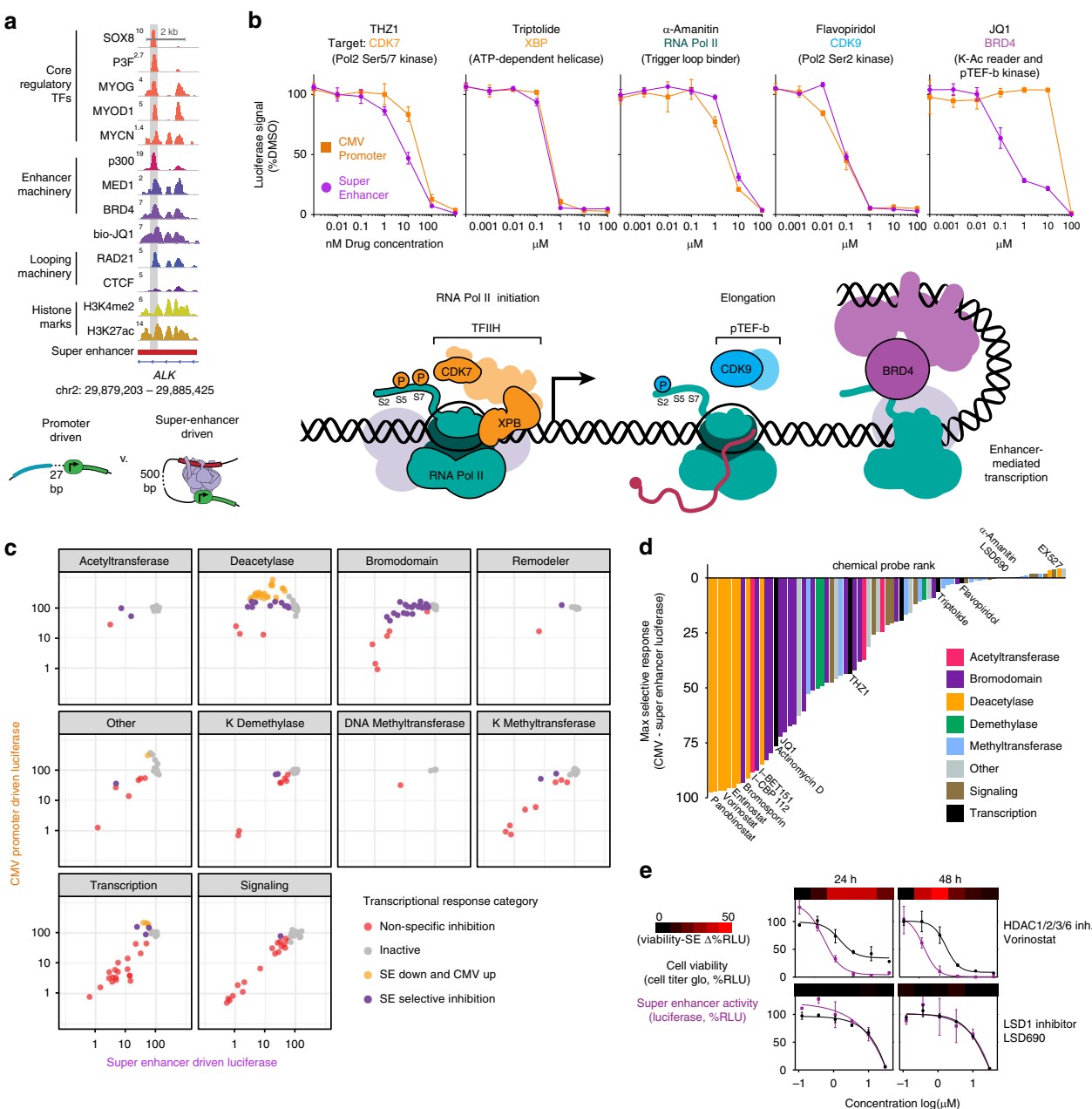

**Fig. 1** Chemical genomics of core regulatory TF-driven transcription. **a** Genetic regulatory element activated by CR TFs at the *ALK* SE in FP-RMS. CR TF-binding region highlighted in gray was cloned upstream of luciferase into a lentiviral expression vector for high-throughput chemical screening. **b** Chemical benchmarking of CMV-promoter driven transcription vs. super enhancer-driven transcription, by dose-dependent inhibition of luciferase expression in FP-RMS cells (RH4). Compounds shown are illustrative of key steps in RNA-Pol2 transcription: initiation (CDK7, XBP dependent), elongation (CDK9 dependent), and enhancer-mediated (BRD4 dependent). Measurements are mean and standard deviation of four technical replicates. **c** Scatter plots of chemical probe selectivity among target classes. Color representative of four transcriptional response categories as shown: non-specific inhibition (high activity against both constructs), inactive, SE-down regulating which also increase CMV-promoter-driven transcription, and SE selective inhibition. **d** Maximum SE selectivity per compound, rank ordered. Mechanistic classes of compounds are distinguished by bar color as shown. **e** Overlay of SE-dependent transcriptional response with cell viability at 24 and 48 h of drug exposure for a SE-selective inhibitor (HDAC inhibitor Vorinostat) and a transcriptionally unselective compound (LSD1 inhibitor LSD690). Heatmap above dose–response curves is calculated as the difference between SE-transcription and cell viability. Points represent mean and standard deviation of three technical replicates

epoxide-bearing cyclic tetrapeptide (Fig. 2c) with structural similarity to trapoxin, the original HDAC inhibitor leveraged to first identify mammalian HDACs[13].

The function of HDACs at active genes or enhancers remains poorly understood, as HDACs are primarily studied through the lens of their role in repressing transcription[14]. Since the goal of the assay was to identify druggable SE dependencies, an on-target

effect of N1302 would predict nuclear-localized HDAC isoforms, such as HDAC2, binding to SEs. Therefore, we sought to define the localization of HDAC2 in relation to SEs and CR TFs in the same FP-RMS cell line used for screening (RH4). We observed binding of HDAC2 within 99% of SEs (Fig. 2d), which co-occupied sites of CR TF binding in SEs (Fig. 2e). HDAC2 was bound predominantly distal (>5 kb from the transcriptional start

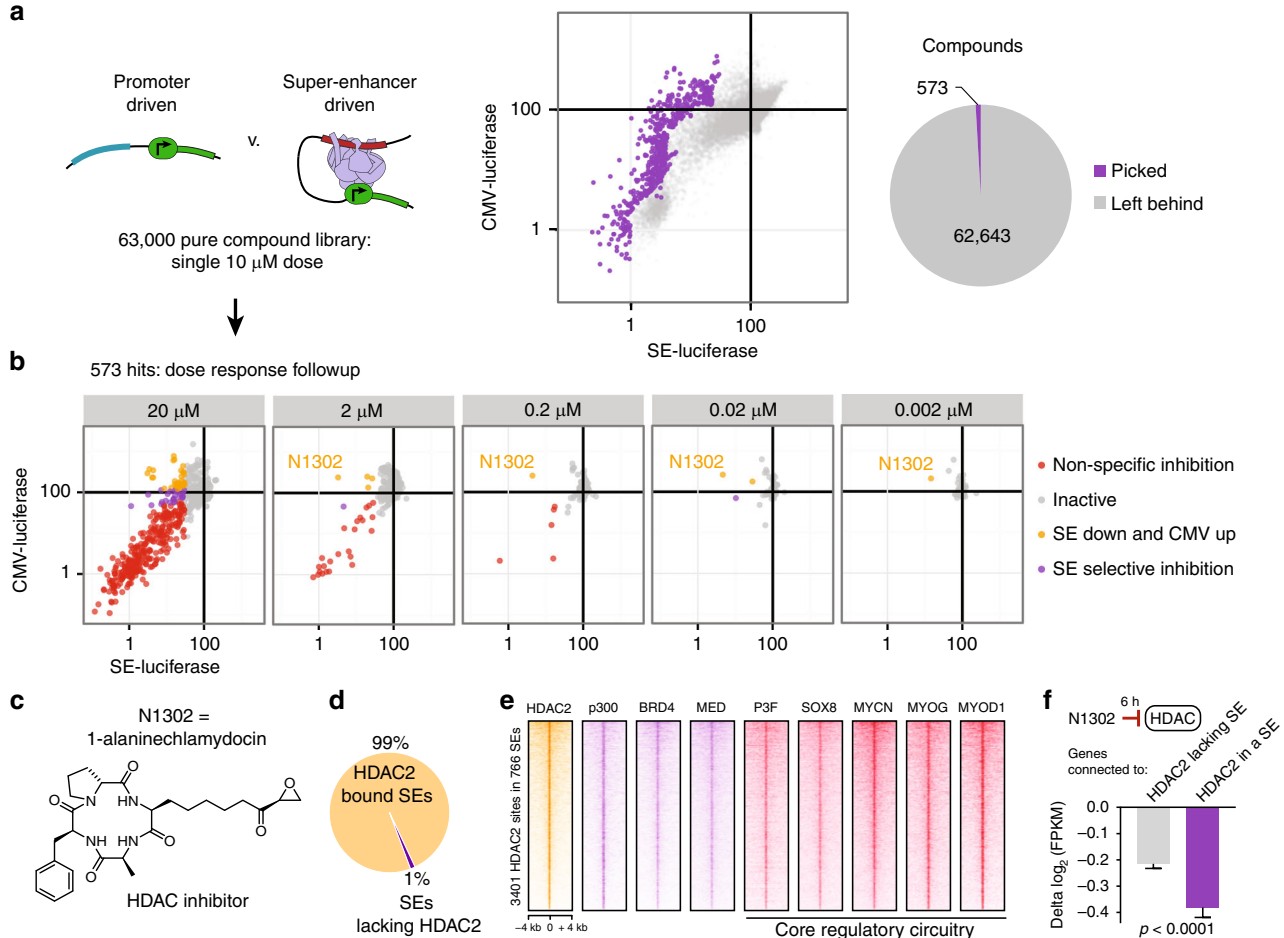

**Fig. 2** HDAC is a key chemical vulnerability core regulatory TF-driven transcription. **a** First pass screening of 63,000 pure compounds against constitutive promoter-driven (CMV) and SE-driven luciferase. Assay was performed after 24 h of 10 μM exposure to each drug in duplicate. 573 of the most selective small molecules were chosen for follow-up screening. **b** Validation of 573 hits by dose–response screening identified the molecules that continued to perform as SE-selective transcriptional inhibitors at lower doses, most notably compound N1302. Points represent the mean of four technical replicates. **c** Structure of N1302, 1-alaninechlamydocin, an epoxide-containing cyclic tetrapeptide known to potently inhibit histone deacetylases. **d** HDAC2 occupies virtually all super enhancers. Pie chart depicts super enhancers in RH4 cells either bound (99%) or unbound (1%) by HDAC2 assessed by overlap of high-confidence ($q < 10^{-9}$) ChIP-seq peaks. **e** HDAC2 sites ($n = 3401$) in SEs ($n = 766$) are co-bound by enzymatically opposing HAT (p300), BRD4, mediator, and core regulatory TFs. ChIP-seq signal is plotted as heatmaps of 8 kb surrounding each HDAC2 peak, as detected in FP-RMS (RH4 cells). **f** HDAC inhibitor has selective transcriptional impact on SE-associated, HDAC2 proximal genes. N1302 was dosed in RH4 cells for 6 h at 1 μM. Error bars represent 95% confidence interval; P-value was calculated by Welch's unpaired t-test

site), was enriched within SEs, and bound with greater intensity at HDAC peaks with SEs compared to regular enhancers (Supplementary Fig. 2a–c). Motif analysis of HDAC2 revealed binding to myogenic E-box in rhabdomyosarcoma cells (Supplementary Fig. 2c), and HDAC2 was found asymmetrically loaded at enhancers of CR TFs (Supplementary Fig. 2e). Furthermore, genes associated with HDAC2 at SEs were downregulated more rapidly by NS1302 than genes associated with HDAC2 outside of SEs, as measured by RNA-seq at 6 h of NS1302 treatment in RH4 cells (Fig. 2f, $P < 0.0001$, Student's t-test).

**Chemical phylogenetic dependencies of HDACs.** To further dissect the contribution of HDACs to the growth of FP-RMS cancer cells, we considered the chemical phylogenetics of HDAC isoforms[15,16]. HDACs are subdivided in the first instance by their dependence on either NAD (Class III, the sirtuins) or Zinc (Class I, IIa/b, and IV) to catalyze the removal of the acetyl moiety from the nitrogen of lysine residues (Fig. 3a). Class I HDAC enzymes (HDAC1, 2, 3, and 8) are closely related structurally and function primarily in the nucleus, linking them to regulation of histone

acetylation states in the context of transcription. Class I HDACs have been reported to be critical for FP-RMS growth[17,18], although this activity has not previously been attributed to disruption of SE circuitry. Indeed, this class was a significant vulnerability (Fig. 3b, $P$-value $= 0.0018$, Student's t-test) for FP-RMS cell proliferation as assayed by CRISPR screening from the Broad's Project Achilles[19]. Using a panel of HDAC inhibitors with diverse and well-characterized isoform selectivity spanning all HDAC Classes (Fig. 3c), we profiled their activity across both FP-RMS and FN-RMS cancer cell lines by imaging cell confluence over time, in concentrations capturing the full dose response. In agreement with the genetic functional screening, Class I HDAC inhibitors were the most potent, whereas little anticancer effect was seen upon inhibition of Class IIa (HDAC4/5/7/9), Class IIb (HDAC6/10), or Class III (Sirtuin inhibitor Selisistat). Among Class I, inhibition of HDAC8 with OJI-1[20] was ineffectual, and those inhibiting HDACs 1/2/3 were most potent (Fig. 3c, Supplementary Fig. 3). The HDAC1/2 selective benzamide Merck60[21] was as potent as HDAC1/2/3 inhibitor Entinostat, yet this was only seen after prolonged exposure (more than 4 days, Fig. 3d, e).

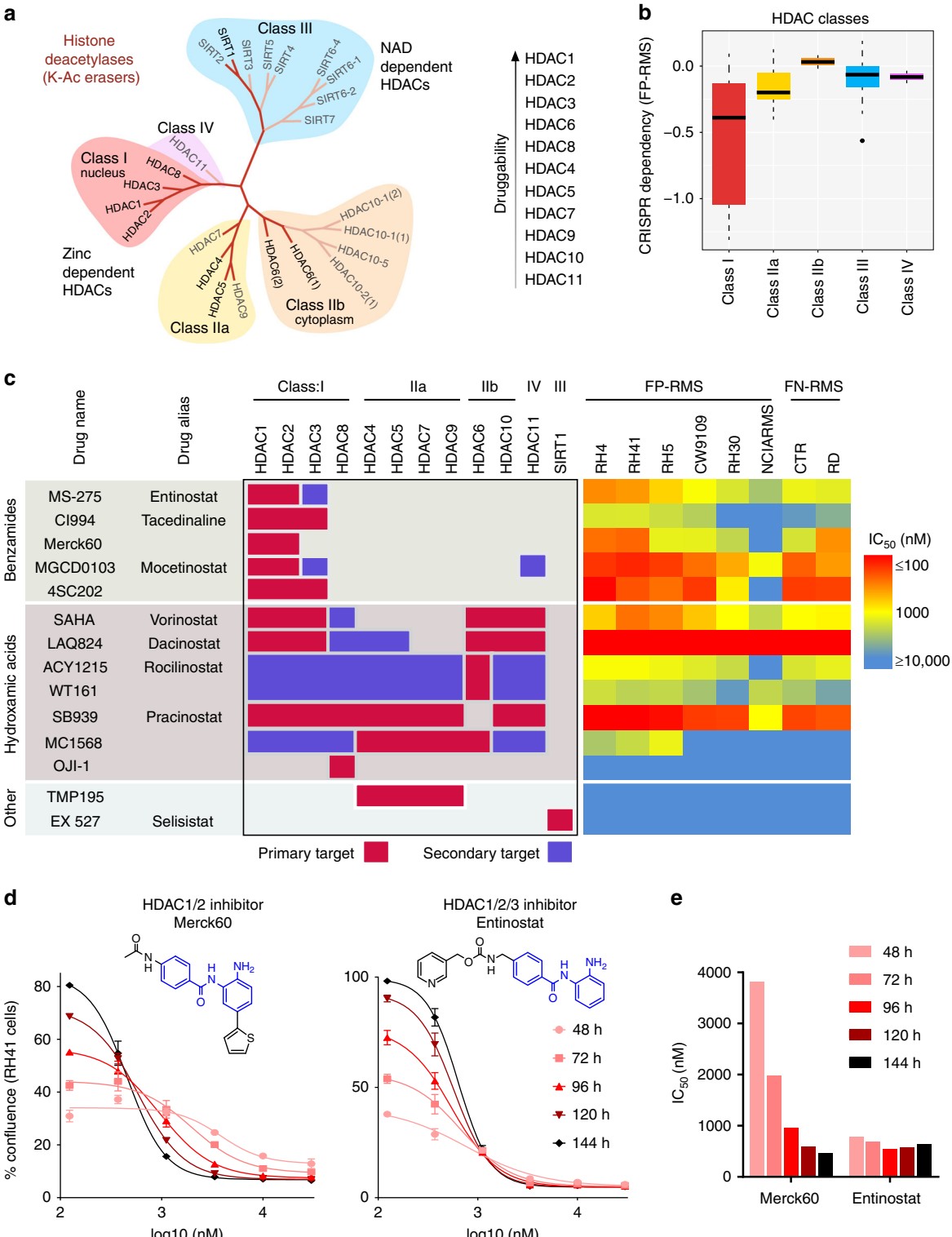

**Fig. 3** Chemical phylogenetics of HDAC sensitivity. **a** Phylogenetic tree of histone deacetylases. **b** CRISPR screening of HDAC isoforms in FP-RMS cells from Broad Achilles dataset reveals strongest dependency on Class I. The panel shows summary statistics as a box (quartiles) and whisker (1.5*inter-quartile range) plot. **c** Diverse HDAC inhibitors with varying selectivity for HDAC isoforms, shown on the left, with each compounds concentration of half-maximal efficacy ($IC_{50}$) on the right (measured by impact on cell growth over time). **d** Dose–response curves of RH41 cell growth impaired by benzamide-based inhibitors of HDAC1/2 (Merck60) and HDAC1/2/3 (Entinostat) over time. Points represent mean and error bars show SEM of triplicate measurements. **e** Rapid stabilization of $IC_{50}$ from Entinostat, compared to gradual decline in $IC_{50}$ with Merck60, corresponding to RH41 cell growth data presented in **b**

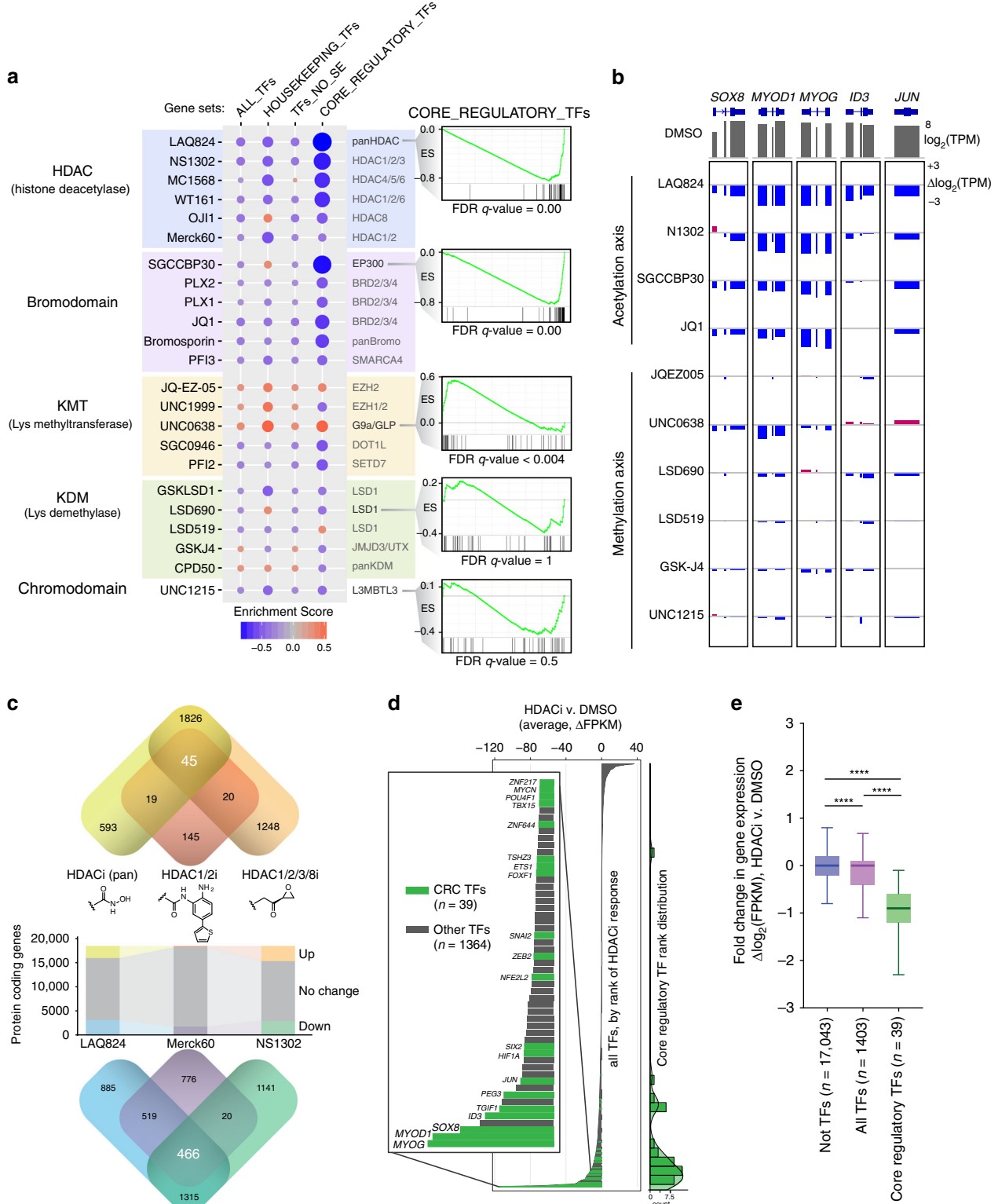

**CR TFs require the acetylation-axis.** To confirm our assays of SE selectivity from a single transcriptional output across tens of thousands of drugs, we sought to move to genome-wide transcriptional analysis with a mechanistically informative set of 23 compounds with various epigenetic targets (Fig. 4a). RNA-seq after 6 h of treatment, followed by Gene Set Enrichment Analysis (GSEA) of various TF categories revealed a selective down-regulation of SE-driven (CR) TFs, especially by chemical perturbation of the histone acetylation-axis (Fig. 4a, Supplementary

Fig. 4a). This same selectivity trend was observed in our *ALK* SE luciferase assay. The bromodomains of acetylation-writer p300 (inhibitor SGCCBP30) and acetylation-reader BRD4 (inhibitor JQ1) were indispensable for SE TFs to remain actively transcribed even on this brief timescale (Fig. 4a). Additionally, HDAC inhibitors were among the most potent at inhibiting CR TFs (Fig. 4a, Supplementary Fig. 4b), such as *SOX8*, *MYOD1*, and *MYOG*, among others (Fig. 4b–d). Interestingly, hydroxamic acid-based pan HDAC inhibitor LAQ824 (Dacinostat), benzamide-based

**Fig. 4** Transcriptome-wide impact of epigenetic probes on core regulatory TF transcription. **a** Chemical informer set of epigenetic probes reveals selective effect on transcription of core regulatory TF genes associated with super enhancers, as compared to all TFs, housekeeping TFs, and TFs not SE-associated. Gene set enrichment analysis was performed on RNA-seq measured gene-level transcripts from 6 h treatments with indicated compounds. Size and color of circles are proportional to GSEA enrichment scores. Mechanistic class is indicated to the left of compound names, whose particular protein isoform targets are indicated on the right. Called out example enrichment plots for CR TFs show strong negative enrichment for pan-isoform HDAC inhibitor LAQ824 and inhibitor of p300 bromodomains SGCCBP30 (with false discovery rates < 0.0001). **b** Suppression of RNA expression of core regulatory TFs upon inhibition of readers, writers, and erasers of the acetylation axis (HDAC, p300 or BRD4), but not by lysine methylation axis (EZH2, G9a, LSD1, JMJD3, or L3MBTL3). Exon level expression changes were quantified from RNA-seq after 6 h of drug treatment, compared to DMSO controls. **c** Changes in protein-coding genes upon HDAC inhibition with three distinct pharmacophores: hydroxamic acid (LAQ824), benzamide (Merck60), and epoxide (NS1302) zinc-binding groups. Overlap in gene sets for increased and decreased genes are shown above and below, respectively. **d** Rank order of change in TF gene expression upon HDAC inhibition, with core regulatory TFs highlighted in green. Values are the average delta FPKM across LAQ824, Merck60, and NS1302. Count histogram of CR TF rank is graphed on the right. **e** Core regulatory TFs are significantly more sensitive to HDAC inhibitors than other TFs, but all TFs (including CR TFs) are also more sensitive than all non-TF genes. Box plots (center line = median, box bounds = quartiles, whiskers = 1.5*inter-quartile range) show $\log_2$-fold change in HDACi versus control DMSO. ****$P$-value < 0.0001 as calculated by a two-tailed $t$-test with Welch's correction

HDAC1/2-selective inhibitor Merck60[21], and epoxide-based HDAC1/2/3/8 inhibitor NS1302 (1-alaninechlamydocin) had negligible overlap in upregulated genes ($n = 45$) and none of which included SE target genes. In contrast, these distinct pharmacophores had substantial overlap with downregulated genes ($n = 466$), which included many SE-regulated genes and CR TFs. Indeed, among all TFs impacted by HDAC inhibition, CR TFs were consistently the most effected, and almost exclusively exhibited a down regulation and not an increase in expression (Fig. 4d, e).

**CR transcription is tri-dependent on HDAC1/2/3.** Next, we developed a chemical probe set to dissect the transcriptional roles of HDAC1 and 2 (of the NuRD complex)[22] and HDAC3 (complexed with N-CoR/SMRT)[23]. Class I HDACs 1 and 2 share the closest homology (85% identical sequence) and can co-compensate for the loss of either isoform, while HDAC3 is the next nearest phylogenetic neighbor (Fig. 3a) with structural variation in the 14 Å pocket adjacent to the Zinc-containing active site[24]. To tune up the selectivity of the benzamide toward HDAC3, we modified the benzamide core, and created a selective HDAC3 inhibitor LW3 based on an analysis of HDAC1, 2, and 3 crystal structures[25]. LW3, a close analog of Merck60 with removal of *para*-thieno ring and introduction of fluorine at *meta* position, largely improved selectivity against HDAC3 over HDAC1/2. Molecular dynamics (MD) simulations of models of Merck60 in HDAC2 and LW3 in HDAC3, respectively, matched with our observation, and suggested that the removal of thieno ring and introduction of fluorine at *meta* position of benzamide core produced shape complementarity of LW3 with the smaller HDAC3 side pocket and improved the selectivity against HDAC3 over other two family members HDAC1/2, which has larger side pocket that can fit thieno ring in Merck 60 (Fig. 5b, Supplementary Fig. 6). Synthetic access to the fluorinated benzamide LW3 was achieved by convergent synthesis (see Supplementary Methods for total synthesis). We utilized an amide bond formation to connect the solvent exposed portion of LW3 **1-1**. After esterification, the resulting acid **1-2** coupled with meta-fluoro, Boc protected aromatic analine to produce the compound **3-1**, which was converted to the final product LW3 after Boc protecting group removal under the acidic condition (Supplementary Fig. 8). Using a biochemical isoform inhibition assay cassette against HDAC1–9 that measures the fluorogenic release of 7-amino-4-methylcoumarin initiated by HDAC enzymatic activity[15], we measured the isoform selectivity for LW3 in parallel with known HDAC1/2 selective inhibitor Merck60 and HDAC1/2/3 inhibitor Entinostat (Fig. 5d). LW3 exhibited inhibitory activity against HDAC3 at 50 nM, with much less activity against

HDAC1/2 at 1600 nM, and no activity against six other HDAC isoforms (Fig. 5d). More importantly, LW3 was able to selectively downregulate SE-driven transcription in a similar fashion to other benzamide-based HDAC inhibitors (Fig. 5c). To demonstrate on-target activity for LW3, we evaluated its impact on the lysine substrate of HDAC3 (pan-H3K acetylation) by western blot, in dose response. An increase in acetylation was observed in dose-dependent manner upon LW3 treatment, and had an effect similar to the increased acetylation created by 48 h disruption of HDAC3 using CRISPR-cas9 and 2 simultaneously delivered sgRNAs targeting the deacetylase enzyme pocket (Fig. 5e).

The HDAC3 selective inhibitor LW3 enabled us to further dissect the relative contribution of HDAC1/2 and HDAC3 to CR TF transcription, which was not reported previously. RNA-seq of treated cells with Merck 60 and LW3 for 6 h revealed that generally, HDAC1/2-selective inhibition had an insignificant impact on all CR TFs, yet did reduce transcription of the three most highly expressed members *SOX8*, *MYOD1*, and *MYCN* (Fig. 5f). HDAC3-selective inhibition with LW3, on the other hand, resulted in selective transcriptional reduction for CR genes. The triple inhibition of HDAC1, 2, and 3 with Entinostat resulted in the strongest downregulation of CR TFs, suggesting that all three of these, but not other HDAC isoforms, are co-essential for CR TF transcription (Fig. 5f, Supplementary Fig. 6a–d). For instance, LW3 inhibits many of the CR TFs, but a central TF MYOG is not affected, and is only depleted upon co-inhibition of HDACs 1, 2, and 3 by Entinostat, or by co-administration of Merck60 and LW3 simultaneously (Fig. 5f). This incomplete impact of inhibiting only HDAC3 is reflected phenotypically in the reduced ability of LW3 to slow FP-RMS cell growth (Supplementary Fig. 6e).

**SE looping defects are induced by HDACi.** The tri-requirement of HDAC1, 2, and 3 suggested to us that, in FP-RMS cells, they may be similarly binding to the epigenome at CR TF locations, and in SEs. To validate this we performed ChIP-seq of all three isoforms individually, and analyzed their binding profiles genome wide, finding a near complete co-occurrence of HDAC1, HDAC2, and HDAC3 at locations bound by CR TFs, with most consistent overlap at CR TF sites in SEs (Fig. 5g). Since the inhibition of all three co-bound HDACs should increase acetylation, HDAC1–3 inhibition should also increase chromatin accessibility at CR TF sites. Thus, we performed ATAC-seq on RH4 cells after 1 or 6 h of HDAC1/2/3 inhibition with Entinostat, and observed that indeed, sites in SEs that are bound by all five CR TFs (MYOD1, MYOG, SOX8, PAX3-FOXO1, and MYCN), which are co-bound by HDACs (bioinformatically defined by HDAC2 ChIP-seq as a representative for all three), underwent a dramatic increase in

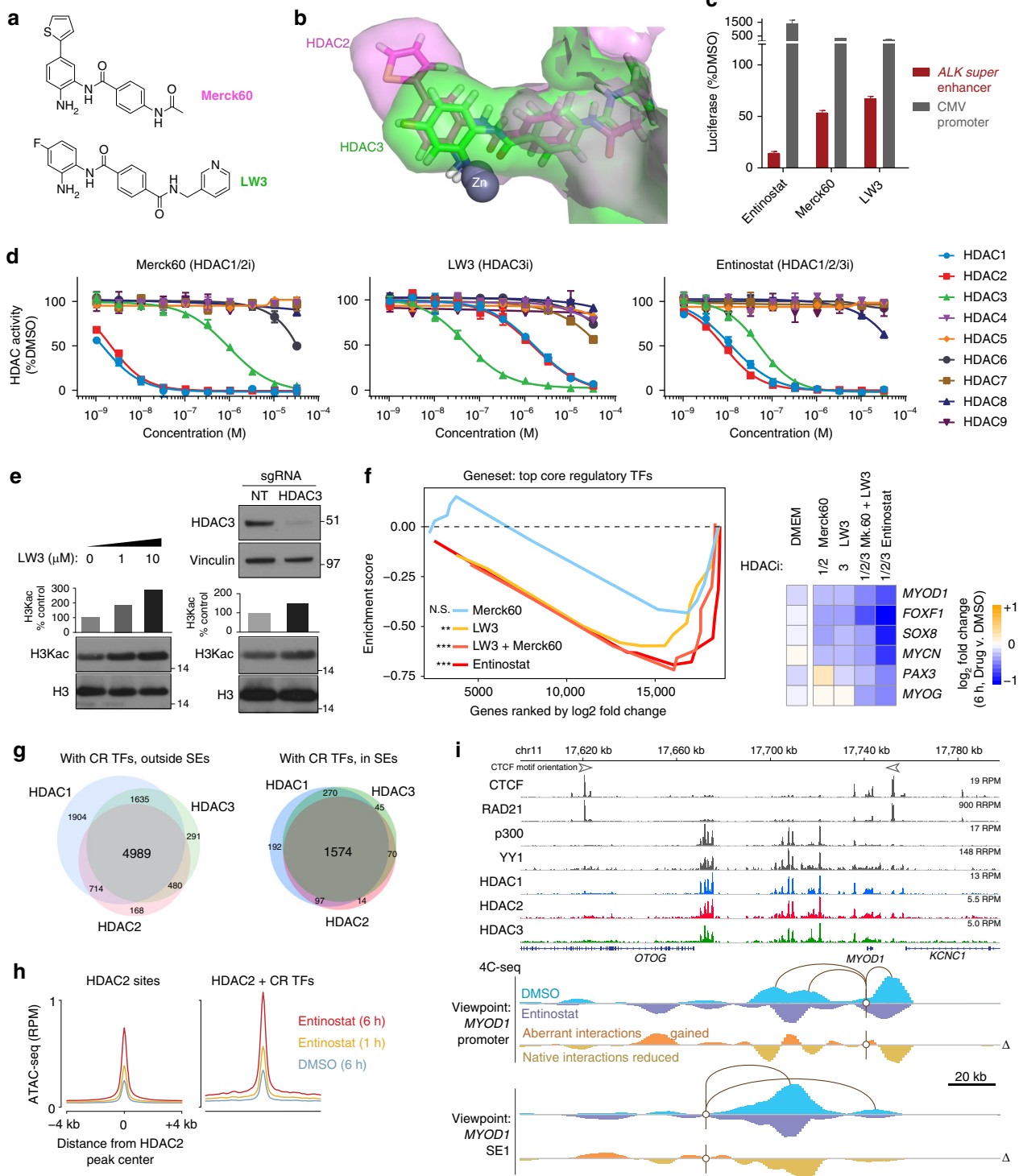

local accessibility (Fig. 5h). On the other hand, sites of HDAC binding outside these CR TF hotspots were opened, but to a lesser extent.

Histone acetylation can increase intra-chromatin pore size from 20 to 60–100 nm[26]. In the interphase nucleus, a SE-contained topological domain of 100 kb (such as at *MYOD1*, distance calculated between antiparallel CTCF ChIP-seq boundary peaks is 130 kb) would be expected to fill a volume of 0.4 μm³ with 15–20% chromatin by volume[27], and be ~130 nm in diameter with a pore size of 60–80 nm. In such an instance, an increase in pore size to 100 nm would triple the volume and dilute

chromatin-associated factor concentrations. Aberrant new contacts would increase, while endogenous, transcription-supportive interactions would be diluted. To test the model that hyperacetylation and induced accessibility opposes normal enhancer–promoter interactions at SE-regulated TFs, we applied circularized chromatin conformation capture (4C-seq) to both the promoter and the first SE of *MYOD1*. In RH4 cells treated with DMSO, the *MYOD1* promoter maintained strong contacts with the central SE components, as well as interactions with the nearby downstream CTCF insulator (Fig. 5g). Upon treatment with Entinostat, these native contacts were the most reduced,

**Fig. 5** HDAC isoforms 1, 2 and 3 are co-essential for complete ablation of CR TF functionality. **a** Chemical structures of LW3 and Merck60. **b** Overlay of MD model of LW3 in HDAC3 (green) and Merck60 in HDAC2 (magenta). **c** Super-enhancer selective transcriptional assay shows LW3 is selective for SE activity, compared to CMV-driven transcription. Experiments were performed in quadruplicate, and error bars represent the standard deviation. **d** HDAC inhibition across isoforms HDAC1–9 reveals selectivity profiles for Merck60, LW3, and Entinostat. Experiments were performed in duplicate for each concentration, with symbols representing the mean and the error bars representing the standard deviation. **e** On-target effect of LW3 observed at the HDAC substrate (acetylated histone 3 lysine residues), seen by western blot after 6 h of treatment at 0, 1, and 10 μM of LW3 (left), and phenocopied by genetic disruption of HDAC3 by CRISPR-cas9 KO at 48 h (right). **f** Core regulatory TF transcription (GSEA, left) is selectively halted by LW3 and Entinostat but not Merck60 (at 6 h in RH4 cells, compared to DMSO). Combination of Merck60 (HDAC1 + 2i) and LW3 (HDAC3i) mimics the strong effect of triple HDAC1 + 2 + 3 inhibitor Entinostat. Expression changes are shown for top CR TFs, for all three benzamide HDAC inhibitors (right). NS = not significant, **$P < 0.008$, ***$P < 0.0001$, determined by GSEA. **g** HDAC1, HDAC2, and HDAC3 ChIP-seq peaks overlap with one another at sites of CR TF binding, with greater frequency of all three HDACs overlapping in SEs, shown summarized by Venn diagrams. **h** ATAC-seq shows increased accessibility to the chromatin template at HDAC2-bound sites, at 1 and 6 h of treatment with Entinostat. Sites are divided into HDAC2-only sites (left) and HDAC2 sites co-bound by all five top core regulatory TFs (SOX8, MYCN, PAX3-FOXO1, MYOG, and MYOD) as measured by ChIP-seq (right). **i** Aberrant chromatin looping interactions gained, and native interactions lost, upon HDAC inhibition with Entinostat for 6 h, assayed by 4C-seq from the MYOD1 promoter (middle) and the first MYOD1 super enhancer (bottom), with ChIP-seq tracks of CTCF, cohesin (RAD21), p300, YY1, HDAC1, HDAC2, and HDAC3

while aberrant contacts were generated. Similarly, 4C-seq from a viewpoint anchor in the first SE of *MYOD1*, which had high contact frequency with the central SE and the *MYOD1* promoter in DMSO-treated cells, lost contact with these sites and gained aberrant interactions nearby and upstream (Fig. 5g). Yet, interaction gains did not traverse beyond CTCF boundaries. This *MYOD1* regulatory domain has large deposits of HDAC1, HDAC2, HDAC3, p300, and loop-mediating factor YY1 bound all in the same pattern, suggesting an intimate connection between HDACs, HATs, and SE loop formation. As proper SE-to-SE and SE-to-promoter interaction is especially required at CR TF genes, we propose chemical-induced loss of contact explains the disabled transcription of these genes.

## Discussion

CR transcription is central to multicellular biology. This lineage-specifying circuitry is hijacked to become oncogenic-circuitry in cancer. Drugging transcription of CR TFs may be more tractable in principle than direct inhibition of TF proteins in many disease states. Indeed, our data here suggests that transcriptional depletion of CR TFs underlies the effect of many epigenetic chemical probes, especially along the histone acetylation axis (BET proteins, HATs, and HDACs).

We have shown here, using chemical genomics, that Class I HDACs (1, 2, and 3 but not HDAC8) are essential to CR transcription and cell proliferation. The auto-regulatory nature of CR TFs makes their universal depletion via chemical perturbation a self-reinforcing collapse. This likely contributes to why CR TFs have the most complete depletion. As this CR collapse is seen by inhibiting lysine acetylation writers (CPB/p300), erasers (HDACs), or readers (BRD2/3/4), these could be thought of as three distinct mechanisms: acetylation constriction (HATi), acetylation bloating (HDACi), or acetylation mimic-distraction (BRDi). As HDACi and HATi have the same (and not reverse) impact on CR transcription, we consider that both are required to strike a balance of acetylation state, which agrees with the observation that p300 and HDACs occupy the same locations on the epigenome (and, positively correlate)[14].

While NuRD is known as a repressive complex, and is involved in enhancer decommissioning during ESC differentiation[28], in RMS cancer cells HDAC is required for transcription of the most active CR TFs, and shows a greater selectivity for CR TF transcription than other strategies such as BRD4 or CDK7 inhibition. BRD4 itself is capable of ejecting nucleosomes by acetylating H3K122[29], yet we cannot exclude the possibility that ATP-dependent remodelers with acetylation-reader domains (e.g. components of the BAF complex PBTM, BRG, BRD7/9) are responsible for local increases in accessibility. However, HDACi-induced chromatin spreading and increase in nuclear size does not depend on ATP[30]. Thus, we suggest that the altered local electrostatics of hyper-acetylated histones, with an average of 17 positively charged residues neutralized by acetylation per octamer[31], leads to inherently more open nucleosome scaffolds. Our data suggests that this effect paradoxically halts core-regulatory genes which are actively driven by coordinated enhancer looping.

Thus, our findings indicate that the rapid transcriptional depletion of CR TFs is likely a principal component of HDACi antiproliferative action across a variety of cancers in which they have been found to be effective. Clinically, these molecules have suffered from poor pharmacological properties preventing adequate tumor availability, but efforts to overcome these barriers are being pursued[32,33]. Nevertheless, HDAC1/2/3 inhibitor Entinostat is being tested for the first time in children with the model disease studied in this work (rhabdomyosarcoma, https://clinicaltrials.gov/ct2/show/NCT02780804), and we are hopeful that insights offered here may serve to inform biomarker development and interpretation of patient responses in the future. Recent data shows Entinostat impacts FP-RMS growth in vivo, and inhibits PAX3-FOXO1 by a multistep and indirect process (studied at 72 h) via a HDAC3–SMARCA4–miR-27a axis[34]. Our early transcriptional and epigenomic experiments (6 h) with isoform selective HDAC inhibitors reveal an additional mechanism, one perhaps more direct and immediate, but not in conflict with additional downstream consequences of HDACi[34,35]. The results here suggest that biomarkers for on-target HDAC inhibition (also in cancers beyond FP-RMS) could include the suppression of CR transcription.

In summary, we have shown a means to transcriptionally choke CR TFs by inhibition of any node in the axis of acetylation (readers, writers, or erasers). Given recent evidence suggesting SEs involve large liquid–liquid phase-separated condensates[36], we speculate that the dynamic opposition of writers and erasers of all epigenomic marks may regulate both the genomic location of condensates (by histone modification) and possibly the local enrichment (or depletion) of non-histone proteins in a given condensate via biophysical property alterations conferred by post-translational modification.

## Methods

**Cell lines and primary tumors**. Cell lines were tested for mycoplasma within one or two passages of each experiment, and cell line identities were ensured by RNA-seq and genotyping. RH4, RH3, RH5, and RH41 were kind gifts from Dr. Peter Houghton, SCMC from Dr. Janet Shipley, RD, CTR and Birch from Dr. Lee Helman. CRL7250 was obtained from ATCC. Validation was performed by DNA fingerprinting AmpFlSTR Identifiler PCR Amplification Kit (Catalog Number 4322288) by Life Technologies. Cell lines were grown at 5% $CO_2$ and 37 °C in DMEM supplemented with 10% fetal bovine serum (FBS) and pen/strep.

**ChIP-seq and Chem-seq**. ChIP-seq was performed as previously described[37,38]. Briefly, cells or tumor tissue was formaldehyde fixed (1% final concentration) for 14 min, dounce homogenized, pelleted, and resuspended in ChIP buffer with protease inhibitors (Active Motif, Cat# 53040). Then, samples were sheared for 27 cycles (1 cycle = 30 s of sonication, 30 s resting) with the Active Motif EpiShear Probe Sonicator. Sheared chromatin samples were immunoprecipitated overnight at 4 °C with antibodies (sources listed in the Supplementary Table 3) and purified by Agarose beads (Active Motif ChIP-IT Sigh Sensitivity Kit, Cat# 53040). Each ChIP reaction was performed at a final volume of 240 μL, with the following antibody final concentrations: 0.08 μg/μL anti-HDAC1, 0.03 μg/μL anti-SOX8, 0.03 μg/μL anti-H3K27ac, 0.05 μg/μL anti-HDAC1, 0.03 μg/μL anti-YY1. Chem-seq was performed following the same as ChIP-seq samples, with the exception that instead of using Protein A/G Agarose beads to immunoprecipitate, we utilized magnetic Streptavidin Dynabeads (M-280, Thermo, Cat# 11205D) pre-incubated with biotinylated derivative of JQ1 (bio-JQ1) as previously reported[10]. In each experiment, validation of enrichment was assessed using qPCR with the ChIP-IT qPCR Analysis Kit (Active Motif, Cat# 53029) using primers listed in Supplementary Table 2. ChIP-seq or Chem-seq DNA libraries were made with Illumina TruSeq ChIP Library Prep Kit, DNA was size selected with SPRIselect reagent kit (250–300 bp insert fragment size). Libraries were multiplexed and sequenced using NextSeq500 High Output Kit v2 (75 cycles), cat. # FC-404-2005 on an Illumina NextSeq500 machine. 25,000,000–35,000,000 reads were generated for each ChIP-seq and Chem-seq sample.

**Assays of transposase-accessible chromatin (ATAC)-seq library preparation**. ATAC were performed as previously described[39,40]. Briefly, 50,000 cells were isolated, and nuclei were generated by incubating on ice with 500 μL lysis buffer (RSB with 0.1% Tween-20) for 10 min. The resulting nuclei were centrifuged at 500 × g for 10 min, and resuspended in 1X Tagment DNA buffer (Illumina) with 2.5 μL Tagment DNA Enzyme (Illumina) and incubated at 37 °C for 30 min. For each transposition reaction, the volume was 50 μL. The transposition mixtures were quenched with 500 μL PB buffer (Qiagen) and purified by standard protocol with MinElute PCR purification kit. Each ATAC library was amplified with Nextera primers for 16 PCR cycles and purified with Agencourt AMPure XP (Beckman Coulter) to remove excess primers. The resulting ATAC libraries were sequenced with NextSeq500 with paired-end reads.

**Analysis of ChIP-seq, Chem-seq, and ATAC-seq data**. ChIP-seq, Chem-seq, and ATAC-seq reads were aligned to the human genome version hg19 using BWA version 0.7.17, and was visualized in IGV after being extended to the average fragment size and binning to 25 bp using IGVtools-count function. Samples with *Drosophila* spike-in were additionally normalized to reads per million mapped dm3 reads[41]. Peaks were called using MACS2 (version 2.1.1.20160309, https://github.com/taoliu/MACS) using "narrow" mode for all targets reported in this paper, as they form sharp genomic peaks (rather than broad swaths, as is seen for H3K27me3). Parameters for MACS2 usage: [--format BAM --control input.bam --keep-dup all --pvalue 0.0000001]. Regions called as peaks which are known to be spurious mapping artifacts were removed before any further analysis (reference locations for sites black-listed by the ENCODE consortium, https://sites.google.com/site/anshulkundaje/projects/blacklists). Motif analysis was performed on peaks called from MACS2, using findMotifsGenome.pl from HOMER version 4.9.1 (http://homer.ucsd.edu/homer/index.html). SEs were identified using the ROSE2 package (https://github.com/linlabbcm/rose2) employing stitching parameter of 12,500 bp. Pipeline code is available on github (https://github.com/GryderArt/ChIPseqPipe).

**RNA-seq sample preparation and data analysis**. RNA-seq[42] was performed as previously described. Briefly, RNA was extracted from RMS cell lines, as well as RH4 and RH41 treated with drugs (listed in Fig. 4), using the RNeasy mini kit (Qiagen). Poly-A selected RNA libraries were prepared and sequenced on an Illumina HiSeq2000. Reads were aligned to hg19 using STAR version 2.5.3a, and gene expression was calculated as Fragments Per Kilobase of transcript per Million mapped reads (FPKM) and Transcripts per Million (TPM) using RSEM version 1.3.0 and the UCSC reference. For GSEA[43] in Fig. 4, the FPKM values for each protein coding gene were averaged across two replicate experiments and $\log_2$-fold-change was calculated by comparison with quadruplicate DMSO control RNA-seq experiments, followed by rank-ordering. For Fig. 5, GSEA was performed using rank-lists of $\log_2$-fold change in TPM, comparing a single drug concentration to its paired DMSO control RNA-seq experiment. Bubble-plots of enrichment output from GSEA analysis of custom and public gene sets were created in R using custom scripts.

**Small molecule compounds**. All molecules were dissolved in DMSO to a final concentration of 10 mM, and diluted to a final DMSO concentration of <0.03% by volume in DMEM for cell culture experiments. JQ1[44], biotinylated-JQ1[10], Merck60[21], OJI-1[20], WT-161[45], JQEZ-005, Cpd50, LSD519, LSD-690, and GSK-J4 were synthesized as previously described. Bromosporine was provided by Peter Brown of the Structural Genomics Consortium. THZ1 was supplied by Nat Gray (Dana-Farber Cancer Institute). Other small molecule inhibitors (Entinostat,

Panobinostat, OTX015, I-Bet-151, I-Bet-762, and I-Bet-726) were generously supplied by Developmental Therapeutics Program (NCI, NIH). A full list of compounds, their mechanisms, their activity, and their sources are available as Supplementary Table 1.

**Western blotting**. Whole protein cell lysates were obtained by using Pierce RIPA Buffer (89900, Thermo Scientific) while total histone cell extracts were obtained by using EpiQuick Total Histone Extraction Kit (#OP-0006, Epigentek). Protein concentrations were estimated by Pierce BCA Protein Assay Kit (#23225, Thermo Scientific). 40 μg of whole protein lysates and 15 μg of histone extracts were run on NuPage 4–12% BisTris minigels (Invitrogen) and transferred to PVDF membranes using iBlot Dry Blotting System (Life Technologies). Membranes were blocked in 5% nonfat dried milk in TBS-T for 1 h at room temperature and incubated with primary antibodies overnight at 4 °C. Antibodies used include αHistone H3 (acetyl K9 + K14 + K18 + K23 + K27, ab47915 from abcam, used at 1:5000), αHDAC3 (ab7030 from abcam, used at 1 μg/μL), αVinculin (V9264, Sigma Aldrich, used at 1:2000), and HRP secondary antibodies αrabbit (sc-2004, Santa Cruz, used at 1:10,000) and αmouse (sc-2005, Santa Cruz, used at 1:10,000). Membranes were developed using an ECL Western Blotting Detection Reagents (RPN2106, GE Healthcare) and then were striped, rinsed, and re-probed with αHistone H3 (#9715, Cell Signaling, 1:2000).

**MD simulations**. Initial models for the HDAC2 simulations were constructed from the crystal structure of HDAC2 (pdb code 4LY1, 1.57 Å resolution)[46] by manually overlaying Merck60 in place of the benzamide ligand in the crystal structure PyMOL. Initial models for the HDAC3–LW3 complex were constructed using the crystal structure of the HDAC3 (pdb 4A69, 2.06 Å resolution)[47] with LW3 placed in a similar fashion as above. Nonbonded parameters for the $Zn^{2+}$ metal ion were determined using the MCPB.py[48] module in Amber16[49] with the empirical ZAFF parameterization method[50]. The rest of the protein system was treated using the AMBER14SB force field[51]. Ligand geometries were initially optimized in Gaussian09[52] using the B3LYP/6–31g(d) functional and basis set followed by charge optimization using B3LYP/6–31 + g(d,p) and RESP charge fitting in the *antechamber* module in Amber16. The *pmemd* module of Amber16 was used for the MD simulations. Protein-ligand systems were solvated by a truncated octahedron of TIP3P waters which extended at least 15 Å away from the protein. All MD simulation used the particle mesh Ewald (PME) method for treating long-range electrostatic interactions, a 10 Å cutoff for nonbonded van der Waals interactions, and periodic boundary conditions. All hydrogen atom bonds were constrained using the SHAKE algorithm and a time step of 2 fs was used to integrate the equations of motion. The systems were minimized in 1000 step increments that gradually reduced restraints on the atoms. The system was then heated to 300 K over 30 ps, followed by equilibration for 10 ps, followed by an NPT equilibration for 100 ps. The temperature and pressure were maintained using a Langevin thermostat and a Berendsen barostat with isotropic scaling, respectively. Each system was simulated for 25 ns in an NVT ensemble. PDBs were generated using the *cpptraj* module in Amber16 and visualized in PyMOL.

**HDAC isoform inhibition assay**. The inhibitory effect of compounds on HDAC1–HDAC9 function was determined in vitro using an optimized homogenous assay performed in a 384-well plate format[15]. In this assay, recombinant, full-length HDAC protein (HDAC1 100 pg/μL, HDAC2 200 pg/μL, HDAC3 100 pg/μL, HDAC4 0.5 pg/μL, HDAC5 10 pg/μL, HDAC6 350 pg/μL, HDAC7 2 pg/μL, HDAC8 16 pg/μL, HDAC9 20 pg/μL; BPS Biosciences, San Diego, CA, USA) was incubated with inhibitory compound for 3 h, and then fluorophore-conjugated substrates MAZ1600 and MAZ1675 were added at a concentration equivalent to the substrate Km (MAZ1600: 8.9 μM for HDAC1, 10.5 μM for HDAC2, 7.9 μM for HDAC3, and 9.4 μM for HDAC6. MAZ1675: 11.5 μM for HDAC4, 64.7 μM for HDAC5, 29.6 μM for HDAC7, 202.2 μM for HDAC8 and 44.3 μM for HDAC9). Reactions were performed in assay buffer (50 mM HEPES, 100 mM KCl, 0.001% (v/v) Tween 20, 0.05% (w/v) bovine serum albumin, 200 μM TCEP, pH 7.4) and followed for fluorogenic release of 7-amino-4-methylcoumarin from substrate upon deacetylase and trypsin enzymatic activity. Trypsin was present at a final concentration of 50 nM (Worthington Biochemical Corporation). Fluorescence measurements were obtained approximately every 5 min using a multilabel plate reader and plate stacker (Envision, Perkin-Elmer). Data were analyzed on a plate-by-plate basis for the linear range of fluorescence over time. The first derivative of data obtained from the plate capture corresponding to the mid-linear range was imported into analytical software (Spotfire DecisionSite and GraphPad Prism). Replicate experimental data from incubations with inhibitor were normalized to DMSO controls. $IC_{50}$ is determined by logistic regression with unconstrained maximum and minimum values.

**Time-course of dose response cell growth assay**. Dose responses were performed by quantifying percent cell confluence from phase contrast images taken every 4 h using the Incucyte ZOOM in 384-well plate format. Dose response was achieved using a range of 12 concentrations from 30 μM to 0.17 nM (dilutions divided by 3), and were performed in triplicate. Cells were plated to achieve 15% confluence at time of drug dosing, and monitored until control (DMSO) wells

reached >95% confluence. $IC_{50}$ values were calculated for each time point using the R statistical package drc (https://cran.r-project.org/web/packages/drc/drc.pdf).

**Luciferase-expressing cells**. A full description of creation and validation of SE construct was described previously[6]. Briefly, pGreenFire vector (with a minimal CMV promoter insufficient for basal transcription) from Systems Biosciences was modified by insertion of sequence from the CR TF-bound intronic SE within the *ALK* gene (chr2:29880537–29880842 in hg19). Introduction into RH4 cells of the lentiviral vector was done in a pooled cell fashion which were selected for successful insertion of the construct using puromycin.

**High throughput screening of compound libraries**. RH4 cells stably transfected with either the pALK-Luc construct (for reporting on PAX3-FOXO1 activity) or a pCMV-Luc construct (reporter for impact on non-PAX3-FOXO1-related transcription) were enriched for reporter expression levels and used for assay development. A high throughput screening assay was then developed after optimization of cell seeding density, length of incubation of cells prior to, and post treatment with test compounds, and effect of passaging among other factors. The assay was validated for sensitivity and reproducibility over three separate runs using 3000 randomly chosen screening compounds with high correlation (coefficient median $R^2 = 0.87$, range $R^2 = 0.86$–0.90). A pure compound library of 63,000 compounds consisting of synthetic and natural products described previously[53] was used in a high-throughput screening campaign for the identification of inhibitors of PAX3-FOXO1 luciferase reporter activity in Rh4 ALK-Luc cells with minimal effect on cell viability and activity in the Rh4 CMV-Luc reporter. DMSO Solutions of screening library of compounds were thawed and used to prepare 100 mM solutions in growth medium. Rh4 ALK-Luc cells were seeded in both white-walled and bottomed 384-well plates (Perkin Elmer, Cat# 6007658) for luciferase assays and clear 384-well plates (Perkin Elmer, Cat# 6007688) for an XTT viability assay by transferring 27 mL of cell suspensions (seeding 3000 cells per well) into each well and transferred into an incubator for 18–20 h. Rh4 CMV-Luc cells were similarly seeded into white plates and incubated. Screening compounds along with the positive (Actinomycin D, Sigma, Cat# A1410) and negative (DMSO) controls were added by transferring 3 mL of the prepared 100 mM dilutions using an automated liquid handler (Agilent, Bravo). Treated plates were incubated for 24 h and allowed to equilibrate to room temperature for 30 min. SteadyLite Plus luciferase assay reagent powder (PerkinElmer, Cat# 6066759) was reconstituted in its buffer in parallel and was also allowed to equilibrate to room temperature. After transferring 30 mL of the luciferase assay reagent using a liquid handler, plates were further incubated at room temperature for 10 min. Finally, luminescence measurements were carried out using a multilabel microplate reader (BMG, Pherastar FSX) set in luminescence mode. Cell viability was assessed by adding XTT reagent (10 mL per well) in clear plates containing treated Rh4 ALK-Luc cells. XTT treated plates were read using a plate reader in absorbance mode at 450 nm (PerkinElmer, Envision) after further incubation for 1 h to allow cells to metabolize XTT to a colored formazan product. Luminescence and XTT absorbance values were normalized to the average of the negative controls on each plate and were calculated as a percentage of these controls. Performance of the screening assay was routinely monitored via calculation of $Z$-factor[54] with $Z'$ for all plates found to be ≥0.7. A weighted mean of percent-control of the Rh4 ALK-Luc ($L$), Rh4 XTT ($X$), and Rh4 CMV-Luc ($C$) values for each test compounds was determined employing the formula Weighted average $= (4L + 2C + X)/7$. Compounds with weighted averages above 60 were identified as hits (outliers) after preparing a boxplot of weighted averages. This resulted in the identification of 573 compounds as hits.

**Hit confirmation and screening of compounds targeting the epigenome**. Rh4 ALK-Luc and Rh4 CMV-Luc cells were seeded into separate white-walled and bottomed 384-well plates were incubated and treated with screen hits and epigenome compounds. The 10-fold of the highest test concentration of these compounds were prepared from DMSO stock solutions via dilution in medium. Final DMSO concentration in assay wells was 0.2% or less. Ten-fold serial dilutions of test compounds were made using an automated liquid handler and 3 mL of each dilution was added to each well of plates containing Rh4 ALK-Luc and Rh4 CMV-Luc cells in 27 mL medium. Each test concentration was assayed in quadruplicate. Corresponding DMSO dilutions were used as control. Treated plates were then read following the same procedure used for screening described above.

**4C-seq**. Chromatin conformation capture was performed with the in situ modification (i.e. performing ligation within intact nuclei) used in HiC experiments. Fixed cells were exposed to 4-bp cutter DpnII for the primary restriction enzyme, re-ligated, followed by reversal of crosslinks and purification. Csp6I reduced template sizes, followed by a second re-ligation step, prior to inverse PCR for viewpoint amplification. 4C samples of RH4 cells treated with DMSO or Entinostat were amplified at the viewpoints for the SE or promoter, with 4C primers listed in Supplementary Table 2. Resulting products were prepared for sequencing using the same procedure as ChIP-seq DNA products, and sequenced on Illumina NextSeq 500. Importantly, 4C samples were co-sequenced with ChIP-seq in order to prevent sequencing failure due to the low sequence complexity of the inverse PCR 4C reaction.

**Reporting summary**. Further information on research design is available in the Nature Research Reporting Summary linked to this article.

## Data availability

The GEO accession number for ChIP-seq data and RNA-seq reported in this paper is GSE116344. All other relevant data supporting the key findings of this study are available within the article and its Supplementary Information files or from the corresponding authors upon reasonable request. A reporting summary for this Article is available as a Supplementary Information file.

## Code availability

Software and code used herein is available at https://github.com/GryderArt.

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

## Acknowledgements

We would like to thank David Levens, Bob Hawley, and Emma Chory for thoughtful discussions regarding experiments and the manuscript. We are very grateful to Peter Brown of the Structural Genomics Consortium for providing small molecules used in this study (detailed in Supplementary Table 1). This work was supported in part by grants from the National Cancer Institute (NCI) P01-CA066996 and P01-CA142106 (to J.Q.). The content of this publication does not necessarily reflect the views or policies of the Department of Health and Human Services, nor does mention of trade names, commercial products, or organizations imply endorsement by the U.S. Government. S. P. is a recipient of a Fondazione Veronesi fellowship. R. R. is supported by Associazione Italiana Ricerca sul Cancro (AIRC 15312) and Italian Ministry of Health (PE-2013-02355271).

## Author contributions

B.E.G., J.Q., J.F.S and J.K. conceived the project. B.E.G. wrote the manuscript. L.W. synthesized and characterized small molecules used in this study. All authors contributed intellectually to the editing of the manuscript and the interpretation of data. G.M.W. designed, supervised an executed all screening experiments. B.E.G., S.P., P.M.C.P., A.C., B.Z.S. and Y.S. performed wet-lab experiments and generated data. B.E.G. performed all genomic bioinformatic experiments. M.E.Y., J.F.S., J.Q. and J.K. supervised the work and mentored the co-first authors. R.R. supervised S.P.; T.R.Q. and O.W. performed molecular dynamics simulations.

## Additional information

**Competing interests:** All authors declare no competing interests.

