## [Peer Review File · Nature Communications]

Reviewers' comments:

Reviewer #1 (Remarks to the Author):

In this comprehensive study, Khan and colleagues utilize a luciferase-based reporter system for super enhancers (SE) to perform a thorough evaluation of the contribution of various effector proteins in modulating transcription. Screening a large library of small molecules, the authors searched for inhibitors that decrease the luciferase signal, classifying the identified targets as components of SE-dependent transcription. The use of this clever genetic probe led them to (surprisingly) identify HDACs as key regulators. The authors go on to show that HDAC1/2/3 are the isoforms that inhibit transcription with HDAC3 as a selective regulator, to which they develop a specific inhibitor (LW3). The the global analysis and the new inhibitor serve as great resources, however for this paper to be a strong candidate for Nature communications I would like for the authors to address the following points:

1. Probe design:

- a. Authors should clearly state that the probe design was published before.
- b. What is the distance between the SE and the TSS? How was this linker length selected, as these vary in the endogenous setting (reaching millions of bp)?
- c. Does the plasmid-based reporter get incorporated into the host genome? If it stays circular, does that constrain DNA looping between the SE and TSS?
- d. The authors use the classification of SE-selective compounds but do not mention how were these statistically determined? What is the cutoff for reduction in SE-dependent transcription they consider significant (distinguishing it from inactive, for example)?

2. For the initial screen, the authors chose 24 hours incubation, which is very long for a transcription time scale. Were any other time points considered/tested? Also, all further analyses were performed in a 6-hour time point. What is the reason for the deviation from the initial 24 hour one? How does this new time-scale effect the analysis?

3. The authors claim that HDACs, particularly HDAC3, halt transcription by making the TF binding sites hyper-accessible and disrupting chromatin looping, but do not provide a mechanistic explanation as to how less acetylation would induce chromatin opening. Is it due to non-histone substrates? Acetylation sites that participate in recruitment of proteins rather than electrostatic interaction with the DNA? Is it possible that HDAC3 has upstream substrates (or target genes) that their expression/function affect SE? In addition, the authors mention that in a co-submitted Nature Genetics paper they found hyper-acetylation spreading of SE, but how does the presence of HDACs that remove acetylation would promote hyper-acetylation at the same site?

Similarly, the authors mention that p300 and Brd4 were also indispensable for SE TFs to remain actively transcribed. However, p300 is an acetyltransferase and Brd4 is an acetyl-reader and thus have the opposite effect of HDACs. How do the authors reconcile this discrepancy?

4. The authors generate an HDAC3 specific inhibitor, LW3. However, they do not demonstrate that the effect they see in cells in response to LW3 treatment is due to HDAC3 inhibition and is not an off-target effect. One way to address this would be to knock down HDAC3 and apply LW3- cells should be desensitized to its effect. While beyond the scope of this paper, an elegant experiment would be to synthesize a version of LW3 with a click handle and perform a proteomics experiment to identify the entire repertoire of its targets.

5. The authors show the effect of Merck60 and LW3 on SE, but not the combination. Does the combination mimic treatment with the less specific Entinostat?

Minor comments:

1. The title of the paper is ambiguous and does not reflect its major findings regarding HDACs.
2. Line 75, should be lower case A.
3. Figure 1a- it would be better to have an illustration of both the reporter and the control construct (as in Figure 2a, or instead)

Reviewer #2 (Remarks to the Author):

In general, the manuscript is well written, with a clear set of experimental goals and justified conclusions. The key takeaway message is summarized well in the abstract, namely that chemical probes of the acetylation axis, and not the methylation axis, selectively disrupt transcription driven by core regulatory transcription factors with chemical probe experiments suggesting HDAC3 as a key isoform. Delineating the HDAC isoforms responsible for this activity, namely class I HDACs that act on chromatin, provides additional information about mechanism and the team uses a broad spectrum of seq-based approaches to test their hypothesis related to the accessibility of CR TF sites and chromatin architecture. I think the story will be of general interest to a wide audience interested in transcription and chemical biology. The work may also have therapeutic implications, and may suggest combinations to test in cellular rewiring experiments. While I can think of a number of follow on experiments that would be interesting, I strongly think that these are beyond the scope of the current study and would be reluctant to list them here because I do not want the editorial staff to view them as part of the current study or require the authors to go beyond the current report. I am curious to know more about the companion work that is submitted to Nature Genetics but realize that these are two separate studies. The only critique that I have is relatively minor and that is to realize that the field has not entirely embraced the statement about SEs as liquid-liquid phase separated condensates. They are correct to cite this work but I might rephrase to state 'given recent work suggesting that SEs involve...' so as to appropriately cite this important suggested mechanism. In general, the figures are easy to understand and the methods section is of appropriate technical depth. Statistical analyses are appropriate. I think this manuscript requires little in the way of editing prior to publication.

Line by line responses are given in blue, below.

Reviewers' comments:

Reviewer #1 (Remarks to the Author):

In this comprehensive study, Khan and colleagues utilize a luciferase-based reporter system for super enhancers (SE) to perform a thorough evaluation of the contribution of various effector proteins in modulating transcription. Screening a large library of small molecules, the authors searched for inhibitors that decrease the luciferase signal, classifying the identified targets as components of SE-dependent transcription. The use of this clever genetic probe led them to (surprisingly) identify HDACs as key regulators. The authors go on to show that HDAC1/2/3 are the isoforms that inhibit transcription with HDAC3 as a selective regulator, to which they develop a specific inhibitor (LW3). The the global analysis and the new inhibitor serve as great resources, however for this paper to be a strong candidate for Nature communications I would like for the authors to address the following points:

We thank the reviewer for considering the work of value to the larger community, which is our goal. We have further improved our manuscript based on the following constructive comments:

1. Probe design:

a. Authors should clearly state that the probe design was published before.

We have clarified this in the text in page 3.

b. What is the distance between the SE and the TSS? How was this linker length selected, as these vary in the endogenous setting (reaching millions of bp)?

During the cloning and enhancer selection process, 16 different super enhancers were evaluated with different sizes, and from diverse SEs identified in FP-RMS cells. Among various constructs, ALK rose as the most active. Then, we evaluated a 2000 bp distance and a 500 bp distance, and the shorter distance between the TSS and the SE gave stronger and more consistent results. This contrasts with the 27 bp linker between the TSS and the CMV promoter in the counter-screen construct. We added this information in Figure 1a.

c. Does the plasmid-based reporter get incorporated into the host genome? If it stays circular, does that constrain DNA looping between the SE and TSS?

Yes, the functional component of the reported gets integrated since we used Lentivirus and generated stable expressing cell lines. The lentiviral integration occurs at random within the genome and we used a polyclonal population selected for high expression of the fluorescent tag. Thus, this creates a non-circular and importantly a histone-incorporated template for TF binding. Previously, we validated that the specific PAX3-FOXO1 binding motif was essential for this construct to work, and that it responds to knock-down of PAX3-FOXO1, which constituted the main criteria for "transcriptional selectivity" to enable comparison to the CMV promoter driven construct. We apologize for the confusion and clarified this in the manuscript in page 3.

d. The authors use the classification of SE-selective compounds but do not mention how were these statistically determined? What is the cutoff for reduction in SE-dependent transcription they consider significant (distinguishing it from inactive, for example)?

We thank Reviewer for bringing up this very important point. We added a detail description of the cutoffs used for response classification to the main text on page 4. Our cutoff recognized the prior literature findings that JQ1 is selective for SEs, and that alpha amanitin is not, which

further validated and gave confidence that our thresholds can successfully classify these diverse responses.

2. For the initial screen, the authors chose 24 hours incubation, which is very long for a transcription time scale. Were any other time points considered/tested? Also, all further analyses were performed in a 6-hour time point. What is the reason for the deviation from the initial 24 hour one? How does this new time-scale effect the analysis?

The 24-hour time point was chosen empirically by time course studies upon knock-down of PAX3-FOXO1, where we determined our time point based on two important factors: (1) strongest reduction of luciferase in cells containing the SE-driven construct and (2) little cell death as measured by cell titer glo. On the other hand, time course on RNA-seq studies of HDAC inhibitors were conducted (in a separate paper submission as the data is beyond the scope of current study), and the results indicated that while 24 hours showed a very strong reduction in SE-driven transcription, the same effect became visible at 6 hours (but not earlier). Thus, we chose 6 hours rather than 24 hours for RNA-seq studies because we sought to reduce indirect transcriptional effects.

3. The authors claim that HDACs, particularly HDAC3, halt transcription by making the TF binding sites hyper-accessible and disrupting chromatin looping, but do not provide a mechanistic explanation as to how less acetylation would induce chromatin opening.

We apologize to the reviewer for our unclear statement caused confusing paradigm. We also noted that in the abstract, we incorrectly said

- *“We confirmed that HDAC1/2/3 are the isoforms that halt CR transcription...”.*

We have corrected this to say

- *“We confirmed that HDAC1/2/3 are the **co-essential** isoforms that **when inhibited** halt CR transcription...”.*

*We suggest that HDAC **inhibition** will increase accessibility by **increasing** acetylation, not decreasing it. This statement is now supported by a new figure that showing increased histone acetylation upon HDAC inhibition (or HDAC knock out by CRISPR) **Figure 5e**. We hope this figure can help avoiding confusion.*

Is it due to non-histone substrates? Acetylation sites that participate in recruitment of proteins rather than electrostatic interaction with the DNA? Is it possible that HDAC3 has upstream substrates (or target genes) that their expression/function affect SE? In addition, the authors mention that in a co-submitted Nature Genetics paper they found hyper-acetylation spreading of SE, but how does the presence of HDACs that remove acetylation would promote hyper-acetylation at the same site?

Here, we are not suggesting the HDACs promote hyper acetylation, but that inhibition of HDAC's function by small molecules causes hyper acetylation. This matches our statement that the HDAC inhibition will change the balance of the acetylation level described next.

Similarly, the authors mention that p300 and Brd4 were also indispensable for SE TFs to remain actively transcribed. However, p300 is an acetyltransferase and Brd4 is an acetyl-reader and thus have the opposite effect of HDACs. How do the authors reconcile this discrepancy?

We would like to suggest that a balance of acetylation is required for SE-driven transcription. In other words, while acetylation is essential for SE-driven transcription, too-much or too-little acetylation can reduce effectiveness of SE function. BRD4 inhibition is more similar to the “too-little acetylation” half of the phenomenon, as it effectively blinds BRD4 from recognizing acetylation. We have added text to aid the readers in thinking through these diverse mechanisms.

4. The authors generate an HDAC3 specific inhibitor, LW3. However, they do not demonstrate that the effect they see in cells in response to LW3 treatment is due to HDAC3 inhibition and is not an off-target effect. One way to address this would be to knock down HDAC3 and apply LW3- cells should be desensitized to its effect. While beyond the scope of this paper, an elegant experiment would be to synthesize a version of LW3 with a click handle and perform a proteomics experiment to identify the entire repertoire of its targets.

*To address this, we have considered that a primary function of HDAC3 is to remove acetylation from histone lysine residues and sought to test if LW3 can achieve this on-target effect in cells. we conducted HDAC3 knock down using CRISPR-cas9 and compared with LW3 treatment. We observed a dose-dependent increase in pan-H3Kac upon LW3 inhibition in 6 hours, as well as the same increase in cells where we used CRISPR-cas9 targeting HDAC3. This confirmative data was now included in **Figure 5 (Figure 5e)**.*

We appreciate Reviewer's suggestions to try the proteomics approach. We prepared a version of LW3 with a biotinylated handle, as the reviewer suggests, but have not been able to perform proteomics experiments with it in time for this resubmission. We agree with the Reviewer that this study is beyond the scope of the current study, but would be an elegant approach to our future study. We hope to have this reviewer's excellent point propel future mechanistic dissection of the next stage of our study.

5. The authors show the effect of Merck60 and LW3 on SE, but not the combination. Does the combination mimic treatment with the less specific Entinostat?

We thank the reviewer for suggesting this experiment. This combination of Merck60 and LW3 indeed did mimic treatment with the less specific Entinostat across the transcriptome at 6 hours. This is now reported with new RNA-seq data analysis in Figure 5f.

Minor comments:

1. The title of the paper is ambiguous and does not reflect its major findings regarding HDACs.

We appreciate this important comment, and have changed the title to make its main findings more apparent. The new title is: "Chemical Genomics Reveals Histone Deacetylases are Required for Core Regulatory Transcription". We also clarify the chemical genomics concept in our abstract.

2. Line 75, should be lower case A.

Thank you, this is now fixed.

3. Figure 1a- it would be better to have an illustration of both the reporter and the control construct (as in Figure 2a, or instead)

This has been adjusted accordingly.

Reviewer #2 (Remarks to the Author):

In general, the manuscript is well written, with a clear set of experimental goals and justified conclusions. The key takeaway message is summarized well in the abstract, namely that chemical probes of the acetylation axis, and not the methylation axis, selectively disrupt transcription driven by core regulatory transcription factors with chemical probe experiments suggesting HDAC3 as a key isoform. Delineating the HDAC isoforms responsible for this activity, namely class I HDACs that act on chromatin,

provides additional information about mechanism and the team uses a broad spectrum of seq-based approaches to test their hypothesis related to the accessibility of CR TF sites and chromatin architecture. I think the story will be of general interest to a wide audience interested in transcription and chemical biology. The work may also have therapeutic implications, and may suggest combinations to test in cellular rewiring experiments. While I can think of a number of follow on experiments that would be interesting, I strongly think that these are beyond the scope of the current study and would be reluctant to list them here because I do not want the editorial staff to view them as part of the current study or require the authors to go beyond the current report. I am curious to know more about the companion work that is submitted to Nature Genetics but realize that these are two separate studies.

We are very grateful for the Reviewer's overall positive feedback for finding this work well written, clearly executed and well justified in its main conclusions. We are grateful that the manuscript has triggered a variety of follow-on experiments in the reviewer's mind, and while we are curious to know them, we understand the reviewer's reasons to not include them here.

The only critique that I have is relatively minor and that is to realize that the field has not entirely embraced the statement about SEs as liquid-liquid phase separated condensates. They are correct to cite this work but I might rephrase to state 'given recent work suggesting that SEs involve...' so as to appropriately cite this important suggested mechanism.

We thank Reviewer for pointing this out. We have corrected the content to prevent overstatement.

In general, the figures are easy to understand and the methods section is of appropriate technical depth. Statistical analyses are appropriate. I think this manuscript requires little in the way of editing prior to publication.

REVIEWERS' COMMENTS:

Reviewer #1 (Remarks to the Author):

The authors have answered all of my questions and have provided additional explanations/clarifications where required. I am happy to accept this revised version as is and have no further remarks.

Yael David, PhD.